# Cancer Immunotherapy: Where Next?

**DOI:** 10.3390/cancers15082358

**Published:** 2023-04-18

**Authors:** Walter Bodmer, Vita Golubovskaya

**Affiliations:** 1Weatherall Institute of Molecular Medicine, Department of Oncology, University of Oxford, Oxford OX3 9DS, UK; 2Promab Biotechnologies, Richmond, CA 94806, USA; vita.gol@promab.com

**Keywords:** immunotherapy, CAR T cells, bispecific antibody, T cell receptor

## Abstract

**Simple Summary:**

This review highlights the challenges and future directions of cancer immunotherapies. Monoclonal antibody therapies, adaptive immunotherapies, CAR T cells, T cell attracting bispecific antibodies, TCR mimic antibodies, and new targets are reviewed. The bispecific T cell attracting monoclonal antibody-mediated killing of cancer cells may be the most promising approach for achieving novel effective cancer immunotherapies.

**Abstract:**

The fundamental problem of dealing with cancer is that cancer cells are so like normal cells that it is very hard to find differences that can be a basis for treatment without severe side effects. The key to successful cancer immunotherapy will be based on a very careful choice of cancer targets that are sufficiently cancer specific not to cause serious side effects. There are two fundamentally different ways to deploy the immune system for such cancer treatments. One is to increase the efficacy of the cancer patient’s own immune system so that it attacks these differences. This has been achieved by “checkpoint blocking” which is very successful but only with a relatively small proportion of cancers. Secondly, one can produce antibodies, or T cells, whose specificity is directed against proteins expressed differentially in cancers. CART cell treatments have proved very effective for some blood cancers but not so far for common solid tumours. Humanised, unmodified monoclonal antibodies have been used extensively for the treatment of certain adenocarcinomas with modest success. However, using antibodies together with the body’s own immune system to treat cancers by engineering monoclonal antibodies that are directed at both a target antigen on the cancer cell surface and also against T cells shows promise for the development of novel immunotherapies. Genes can be found which are expressed highly in some cancers but with a low or absent expression on normal tissues and so are good novel targets. It is so far, only immune-based killing that can kill bystander target negative cells, which is essential for successful treatment since hardly ever will all the cells in a cancer express any desired target. We conclude that, while there still may be many hurdles in the way, engineered bispecific T cell attracting monoclonal antibody-mediated killing of cancer cells may be the most promising approach for achieving novel effective cancer immunotherapies.

## 1. Introduction

Cancer remains a major cause of death worldwide, with nearly 10 million deaths per year. In spite of major advances in treatment and prevention over the last 20–30 years, the worldwide number of cancer deaths per year is increasing because of an overall increase in the proportion of older people and the effects of Western-style improvements in living conditions in many parts of the world. Smoking, infection, and obesity remain major targets for prevention together with improved early detection, but there is still a huge need for improved cancer therapies, especially for the commoner adenocarcinomas, which account for more than 60% of the total deaths from cancer [1]. In this brief perspective, we provide an overview of current and future directions for cancer immunotherapy, with an emphasis on monoclonal antibody-based therapies.

Dobzhansky, the outstanding evolutionary geneticist, famously said that “Nothing in biology makes sense except in the light of evolution” [2]. This applies to cancer at two levels, namely (1) that cancer is itself an evolutionary process at the somatic level [3], and (2) that since cancer is a disease that has its major effect after the end of the reproductive period, there has been little or no evolutionary pressure to reduce the incidence of cancer. The importance now of cancer in human populations is simply a product of the fact that our conditions of living, together with improved management of diseases, have extended the average life span to an extraordinary extent beyond the end of the reproductive age and well beyond the average human life span during nearly all the period of human evolution.

The genetic mutations and stable epigenetic somatic changes that drive the progress of a cancer are the only potential source of reasonably frequent stable differences between cancer cells and normal cells that can provide the basis for cancer therapies that do not lead to severe side effects. Focusing on the changes that give a significant advantage to the growth of a cancer means attacking changes that are likely to be relatively common in one or more of the many different types of cancers (Table 1).

The fundamental problem, however, of dealing with cancer is that cancer cells are mostly so similar to normal cells that it is very hard to find differences that can be a basis for treatment without damaging some important aspect of normal function.

Major common cancer mutations in genes such as TP53, APC, or Kras have proved very hard to target either with small molecular weight inhibitors or by immunotherapy. Most commonly used small molecular weight drugs for cancer treatments target tyrosine kinases such as EGFR, which are not cancer-specific and so give rise to significant side effects. They do help but are clearly not the ultimate answer to successful specific cancer treatments. Methylation changes are hard to target and mostly result in switching off gene expression that is connected with the control of differentiation. The challenge here is to reverse the lost gene expression and, for example, try to deliver the normal version of the relevant gene to the cancer by gene therapy. However, achieving such targeting to all the cells in a cancer may be very difficult, if not impossible.

An adenocarcinoma will nearly always contain more dividing cells than the normal adjacent tissue because it necessarily contains fewer if any, fully differentiated cells. Thus, any gene involved in any aspect of the control of cell division will seem to be overexpressed in such a cancer, and so is very likely to be associated with a relatively poor prognosis. This is because the less differentiated an adenocarcinoma, the more aggressive it is likely to be. This, in turn, means that most, if not all of the often noted gene expressions associated with a poor cancer prognosis are very unlikely to be good targets for cancer treatment without damaging side effects.

The implication of this brief introduction to general aspects of cancer is that the key to successful cancer immunotherapy will be based on a very careful choice of cancer targets that are sufficiently cancer specific not to cause serious side effects. Deploying the immune system for cancer treatments, therefore, offers the possibility of minimising the problem of side effects.

There are two fundamentally different ways to deploy the immune system. One is to increase the efficacy of the cancer patient’s own immune system for an effective attack on differences between the patient’s cancer and normal cells. The second is to use antibodies, or T and other immune system cells, that are engineered to use the patient’s immune system so that it attacks differences between normal and cancer tissues. This review will focus mainly on T cell-directed immune therapies.

## 2. Adaptive Immune Therapies

Novel cancer proteins produced by mutations in oncogenes such as TP53 and Kras should, in principle, be cancer-specific targets for the patient’s adaptive immune system to attack, but that has been found to be very difficult to implement. That may well be because the immune system has not evolved to deal with such differences since, as discussed above, cancer has not been a major evolutionary driver. It is also not clear that effective antibodies or T cells would be produced against ectopically expressed proteins, namely proteins that are expressed in a cancer but not usually in any but a small set of other normal cells such as the testis or the placenta. Such proteins are not novel to the overall immune system as they are normally expressed in some of the body’s normal cells, and so induce immune tolerance.

The first clear evidence for an immune attack against human cancers was the observation of the lack of expression of HLA Class I proteins on the surface of certain cancer-derived cell lines due to the complete loss of expression of β2 microglobulin [4]. This completely protects the cells from adaptive immune T cell attack against intracellular targets since the recognition by T cells depends on the presence of HLA Class1 on the target cell’s surface in order to present peptides derived from the intracellular target on the surface of the targeted cells. It was shown that the complete loss of expression of β2 microglobulin was relatively common in colorectal cancers that were mismatch repair deficient. This would be expected given the relatively high frequency of novel somatic mutations in these cancers due to the mismatch repair defect leading to strong selection against attack by the immune system on these mutation-bearing cancers [5].

There are certain other cancers that contain a large number of abnormally mutated proteins and so presumably stimulate an adaptive immune response. These are cancers in tissues that are normally highly exposed to mutagens. For example, certain areas of the skin are generally exposed to sunlight whose UV rays cause DNA damage, which causes melanomas, while cells in the lungs of cigarette smokers are exposed to mutagens in tobacco smoke, and this is the major cause of lung cancers. The high incidence of mutations in such cancers called the mutational burden, can be easily identified by genomic DNA sequencing, and classifies them as cancers where there are potentially enough novel mutations to elicit a relatively strong adaptive immune response.

It was discovered by James Allison and Tasuku Honjo that certain antibodies, which block functional exhaustion of T cells mediating adaptive immune responses, could enormously increase the immune response to tumours with high mutational burdens and so lead to dramatically improved therapy for such tumours. For this discovery, Allison and Honjo received the Nobel Prize in 2018 [6]. T cell exhaustion is part of the normal mechanism for preventing overactivity of an immune response. The blocking of this T cell exhaustion by certain antibodies when there is a strong adaptive immune response to the mutational burden in a cancer is referred to as Immune Check Point blocking therapy [7] (Figure 1). The application of this therapy to patients with tumours with mismatch repair deficiencies, which are now known to occur in quite a wide range of cancers [8], was the first US FDA-approved novel therapy based entirely on the genetics of the cancer.

While checkpoint blocking is a dramatic new approach to cancer immunotherapy, it is limited to a relatively small subset of cancers and, of course, also limited by the development of resistance to the therapy due to selection for β2 microglobulin and other mutations that interfere with T cell immune responses.

## 3. Direct Monoclonal Antibody Therapies

Ever since the discovery of mouse monoclonal antibodies (Mabs) against defined antigens by Milstein and Kohler in 1975, for which they received the Nobel Prize, the unique specificity of Mabs suggested their possible use for immunotherapy. However, until the development of engineered humanised antibodies derived from the mouse antibodies by inserting their antigen recognition sequences into a human background antibody, there was no realistic prospect of their use for successful immunotherapy [9]. The first successful Mab therapies were directed against EGFR (Epidermal Growth Factor Receptors) for the treatment of colorectal cancer and against the related growth factor Her-2 (Erb-2) expressed in a proportion of breast cancers. This approach was based on the assumption that such antibodies would block cancer growth by blocking the interaction between the growth factor and its ligand. Subsequently, it became clear that the Mabs also functioned by immune-mediated killing by NK cells stimulated by binding the Fc portion of a Mab attached to cancer cells through their Fcγ3 receptor [10]. Though these therapies have been widely used, their effectiveness is quite variable and not as high as had been hoped. In addition, there were significant associated side effects presumably due to blockage of growth and killing of normal tissue cells.

## 4. CAR T Cells

Given the apparent limitations of enhancing the patient’s own adaptive immune system’s attack against their cancers, another way forward is to produce patient-derived, engineered T cells that would kill their cancer. This can be done by replacing a T cell’s receptor with a monoclonal antibody against any desired target so that it kills cells expressing the desired target rather than killing by recognizing whatever HLA-associated peptide its receptor was directed against. To achieve this, the DNA sequence that codes for the scFv antibody fragment that recognises the desired target antigen is linked to a transmembrane domain, one or two stimulatory domains, for example, CD28 and CD137(4-1BB), and a T cell activating domain, CD3ζ (Figure 2A). This construct is then incorporated into a Lenti virus carrier for delivery to T cells. The T cells are obtained from the patient, as T cells from any other source, unless very well matched for its HLA type, would be rejected as foreign by the patient. Sufficient numbers of T cells can be obtained from the patient by leukapheresis, expanded in culture by IL-2 stimulation, transfected in bulk by the lentivirus carrying the engineered construct, and delivered back to the patient (Figure 2B, [11]).

For certain leukaemias and lymphomas, this CAR T cell therapy has proved to be a remarkably effective treatment using targets such as CD19 for B cell lymphomas [12]. BCMA, a promising target for the treatment of myelomas, is currently still undergoing clinical trials [13]. Many attempts have, however, been made to treat adenocarcinomas by CAR T cell therapy with so far very limited, if any, responses. This may be due to the tumour microenvironment (TME) being unfavourable to the penetration of externally delivered CAR T cells.

There are many possible developments of CAR T cell therapy being investigated, including using different combinations of stimulatory domains, selecting the T cells to be transfected by the lentivirus, using lipid nanoparticles (LNP) carrying the relevant sequence as an mRNA for delivery, and considering ways of limiting in the in vivo life span of the CAR T cells by engineering in a time-limited suicide mechanism. There are also approaches being made to choose “universal” T cell sources that would overcome the problem of immune rejection by a non-patient source of T cells [14].

## 5. T Cell Attracting Bispecific Monoclonal Antibodies—TCBiMabs or TCBs

Another approach to using antibodies together with the body’s own immune system to treat cancers is to engineer monoclonal antibodies that are directed at both a target antigen on the cancer cell surface and T cells by attaching the tumour targeting antibody to an antibody against CD3, which is present on essentially all functional T cells. These bispecific antibodies can then bring T cells sufficiently close to the targeted tumour cells to kill them as if they were recognising, say, a viral component on the cell surface [15]. All the components of such an engineered monoclonal antibody must be humanised, that is, based on a human antibody backbone, to avoid the inevitable unwanted immune responses against the foreignness of a non-human antibody component.

Such engineered bispecific T cell-attracting monoclonal antibodies were first called BITES and were based on only attaching the scFv portion of a tumour targeting antibody to the anti-CD3 (reviewed in [16]). This was done on the assumption that the smaller size of such an engineered antibody would enhance its access to a cancer. It was, however, then realised that the Fc portion of an antibody was required to prevent it from being rapidly degraded in blood, which inevitably limits its access to a cancer. In addition, to prevent unwanted immune activity based on a functional Fc antibody domain, it was necessary to inactivate the Fc function by introducing suitable mutations which did not interfere with the stability of the antibody in the blood when it was delivered IV.

An example of the schematic structure of such an engineered bispecific T cell attracting antibody, TCB, as used in initial clinical trials, is shown in Figure 3A, and its mode of action is illustrated in Figure 3B [17]. In this case, the cancer target was CEACAM5, which is present at higher-than-normal levels on the surface of a significant subset of colorectal adenocarcinomas (CRC). This is illustrated in Figure 3C, which shows the relationship between the level of expression of CEACAM5 in a collection of CRC-derived cell lines and the efficiency of the killing of the cell lines by the anti-CEACAM5 TCB. The cell lines fall into two categories with respect to the relationship between CEACAM5 expression and the extent of TCB killing. Thus, there is little or no killing of those cell lines which have less than around 10,000 molecules of CEACAM5 on the surface of each cell. This threshold provided the basis for the choice of the threshold of CEACAM5 expression on patient CRCs above which treatment by the TCB was justified. Initial clinical trials of the treatment of CRCs by the anti-CEACAM5 TCB in combination with an anti-PDL-1 checkpoint inhibitor have given promising results [18].

## 6. T Cell Receptor Mimic Antibodies—Tcrms

The next way forward is to design specific engineered immune-based attacks against novel or relatively overexpressed determinants in cancers, not only on the surface of the cancer cells but also internal to the cells. This raises the problem that both T cell- and antibody-mediated killing only work for targets on the cell surface.

When a protein is expressed inside the cell, it can only be recognised on the cell surface by peptides derived from it that are expressed on the surface of the cell in association with HLA class I or II gene products. This is part of the normal process by which T cells recognise the foreign expression of, for example, viral products that are only expressed inside a virus-infected cell. It is possible, then, to make antibodies that recognise the combination of a specific protein-derived peptide with a given HLA product. These are called T cell receptor mimic antibodies because they mimic the process by which T cells recognise such products using the T cell receptor (Figure 4A). This is dependent on engineering the production of a single protein that links an HLA Class1 molecule to its β2 microglobulin partner and to the peptide derived from the internally expressed protein, as shown in Figure 4B. The protein can be produced by inserting the DNA sequence that codes for it into a suitable viral or plasmid vector under the control of a strong promoter and transfecting the construct into cells that synthesise proteins efficiently. The protein produced can then be used as an antigen in the usual way to immunise mice and produce monoclonal antibodies that only recognise the HLA/ β2m/peptide combination. Such an approach could be used, in principle, to make antibodies that are specific to the HLA combination with peptides from the mutated forms of the Kras and TP53 proteins that are commonly found in cancers. However, the practical application of this approach to specific immune therapy directed against the mutated Kras and TP53 gene products has proved to be very difficult, just as is the case for spontaneous adaptive immune responses to these differences, as already discussed.

There are, however, some recently published promising preclinical models of Tcrm bispecific T cell attracting monoclonal antibody therapies using certain novel antigen targets [19].

## 7. New Targets

The key to successful immunotherapies using either CAR Ts or TCBs is to have good targets in terms of cancer specificity and good monoclonal antibodies against these targets. Once you have a good antibody, there is then a variety of ways, as we have discussed, by which it can be used for immunotherapy.

As already mentioned, oncogene mutations, such as in Kras or TP53, or novel translocation-based proteins, such as the original bcr/abl from the Chr9/22 translocation commonly found in CML (Chronic Myelogenous Leukaemia), and the subsequently discovered similar translocations in some prostate, lung and other cancers, have proved difficult to develop as CAR T or TCB targets.

The much talked about “neoantigens” that arise by chance in proteins by “neutral” mutations with no positive or negative effect on the growth of a cancer are almost entirely unique to a given patient’s cancer and so require an individual strategy for the treatment of each individual cancer. This is a very demanding and expensive approach to treatment that has so far not proved to be practical.

A summary of presently known types of potential cancer targets for T cell and TCB immunotherapy is given in Table 2. Apart from the mutation or translocation-based targets, which are by their nature-cancer specific, there is the possibility that some post-translational protein modifications, such as unusual glycosylation or phosphorylation, may also be cancer-specific. Products of cancer-causing viruses such as HPV, HBV, and EBV are also cancer specific and, in the case of HPV, have been very successfully used for vaccination against cervical cancer.

Ectopic proteins are proteins that are expressed in subsets of a wide variety of cancers but not in the normal tissues from which the cancers have arisen. Such proteins may often only be expressed in a very small subset of normal cells, for example, in spermatocytes or placental-derived trophoblasts, as is the case for PLAP (Placental Alkaline Phosphatase). In practice, such ectopically expressed proteins in a cancer are effectively cancer-specific.

“Cancer Testis” antigens, such as NYESO and the MAGEs, were discovered through the identification of a family of proteins that were the targets of patients’ T cells against their cancers [20]. They are expressed in a wide variety of different cancers and only otherwise in the testis, and so are also effectively cancer-specific. These proteins seem to be similar to another class of more or less cancer-specific proteins derived from the HERVs (Human Endogenous Retroviruses), which appear to be viruses encoded in the genome that are only sometimes activated in cancers [21]. This unusual cancer activation may also occur for some of the many highly repetitive DNA sequences that account for nearly half of the total human genome. These unusual expressions can be sought by careful analysis of the differences between the mRNA content of cancers and normal tissues.

There is no known effect of these unusually expressed proteins on the growth of the cancer. They cannot, therefore, be targeted, for example, by low molecular weight chemical inhibitors of function or proteins or peptides that block some key function. It is only by using the immune system’s capacity for recognition of foreignness that such unusual expressions can be used as targets for killing the cancer. When the expression is only inside the cell, then it will depend on the production of an effective Tcrm.

## 8. Conclusions

Cellular treatments, notably with CAR T cells, have proven effective for some blood cancers but not so far for common solid tumours such as colorectal and breast cancer [22,23]. They may, however, work for ascites tumours common in ovarian cancer. If ways could be found for making CAR T cell treatments of the common carcinomas effective, then the antibodies or T- cell receptors engineered to be immunologically effective against the cancer targets discussed in the previous section would also become the basis for effective CAR T therapies. However, it seems probable that the greater logistic problems associated with cellular therapies would still favour antibody-based immunotherapies.

While there is not yet definitive evidence for the effectiveness of TCB-based cancer immunotherapies, it still seems likely that they will be the future basis for the most effective anti-cancer therapies.

First of all, it is very important to emphasise that it is so far, only, immune based-killing that can kill bystander target-negative cells. Killing bystander cells is essential for successful treatment since, in no case, will all the cells in a cancer express any desired target. There is good evidence for bystander killing by T cells activated by killing target-carrying cancer cells and then killing target-negative cancer cells in their vicinity [24].

There are still many ways in which current approaches to TCB cancer immunotherapies could be improved. Most obviously, there is the use of combinations with a variety of checkpoint inhibitors and small molecular weight drugs such the tyrosine kinase inhibitors. For intracellular targets, there are ways to improve the choice of peptides which bind given HLA types to improve the specificity and affinity of their HLA binding. There are also ways to improve the affinity of the Tcrms binding their HLA/peptide target. There may also be ways to increase the patient’s cancer expression levels of ectopic and other targets. Another approach could be to counter the inhibitory effect of the tumour microenvironment, for example, by attacking tumour-associated myofibroblasts through their expression of FAP (Fibroblast Activating Protein) [25]. There is also considerable promise in improving the delivery of TCBs to their cancer targets, for example, by loading LNPs (Lipid Nano Particles) with the mRNA coding for the engineered antibody construct, following the success of mRNA-based anti-viral vaccination [26].

We would also like to highlight that in addition to T cell-based immunotherapies, there are other immune cell type immunotherapies such as NK-cell-based therapies, CAR-NK [27], and NK cell-specific engagers [28,29,30] that will be developed in the future.

No single cancer target is ever likely to be enough to completely cure a cancer, whatever the approach to therapy. This is because there are nearly always enough cells in a cancer for at least one resistant mutation to occur that will give rise to selection for resistance to the treatment. The development of resistance to treatment can only be securely overcome if several targets are attacked at the same time. This emphasizes the need to search for good new targets for immunotherapy.

Our conclusion is that, while there still may be many hurdles in the way, engineered bispecific T cell attracting monoclonal antibody-mediated killing of cancer cells, based on targeting ectopic and other more or less cancer-specific expressed proteins, is, so far, the most promising approach for achieving novel effective cancer immunotherapies.

## Figures and Tables

**Figure 1 cancers-15-02358-f001:**
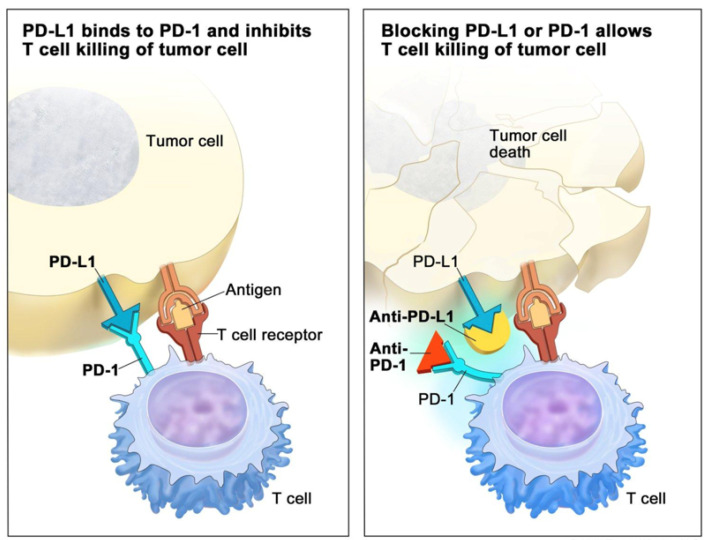
Scheme how checkpoint blockers work. PD-L1 and PD-1 pathway (**left**) and inhibitors are shown (**right**).

**Figure 2 cancers-15-02358-f002:**
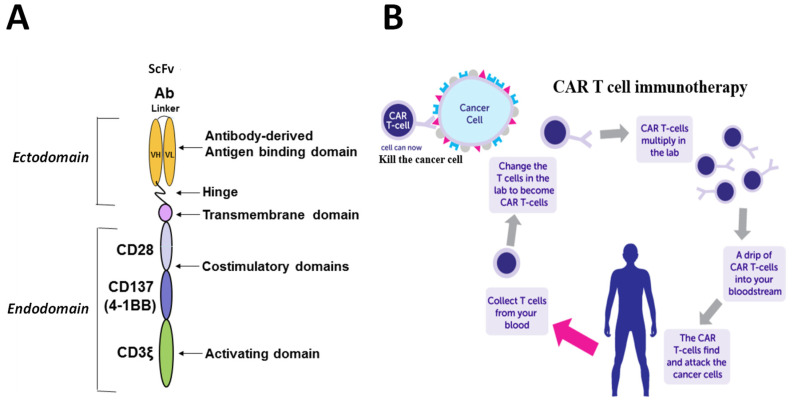
(**A**) CAR T structure. The figure from [11]. (**B**) Scheme of CART cell immunotherapy.

**Figure 3 cancers-15-02358-f003:**
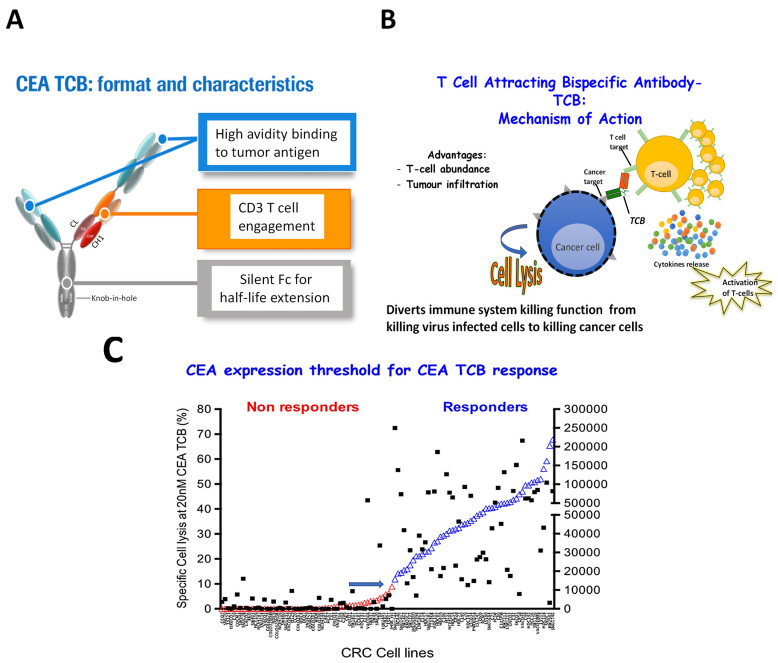
(**A**) T Cell Attracting Bispecific Antibody-TCB: Mechanism of Action. (**A**) The bivalent CrossMab knob-in-hole format of the CEA TCB antibody is shown [17]. (**B**) CEA TCB mechanism of action. (**C**) CEA T cell bi-specific (TCB) response correlates with CEA expression. Two categories of response to TCB-based lysis: Cell lines with above 10,000 surface CEA molecules (blue symbol) nearly all more than 10% lysis. Cell lines with below 10,000 surface CEA molecules (red symbol) are nearly all less than 10% lysis. Right-hand *X*-axis: Blue triangle shows the cell line level of expression responders. Red triangle shows the cell line level of expression non-responders. Left-hand axis; Black square shows the percent of cell lysis of cell line corresponding to the expression level symbol.

**Figure 4 cancers-15-02358-f004:**
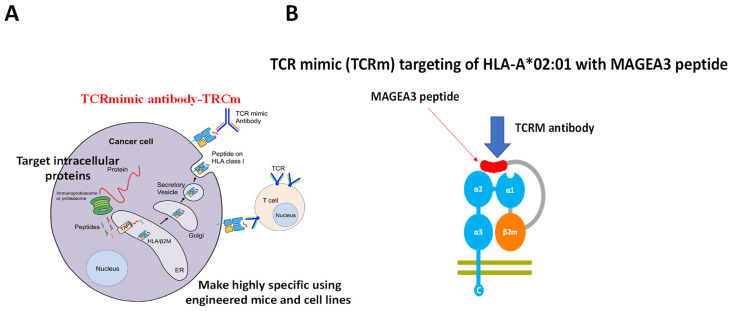
(**A**) TCRm antibody principle. The principle from [19]. (**B**) Tcrm antibody interaction with HLA-2-β2m-peptide peptide construct.

**Table 1 cancers-15-02358-t001:** Mutated and methylated genes found in Colorectal Cancer.

Mutated Genes
Wnt pathway:
APC
CNNBT (beta catenin), CDH1(E-Cadherin), TCF7L2
Apoptosis/DNA repair
TP53
hMSH2
hMSH6 ATM
Cell cycle checkpoints
CDKN2A (p14, p16)
Growth factors and receptors/signaling
SMAD4, Kras
TGFBR2(TGFbetaIIR), Braf, PIK3CA, FBXW7
ERBB2, MAP2K4, Nras, SMAD2
Other
ARID1A, FHIT, ZFP36L2, KMT2C
Immune resistance
B2M(beta2m), HLA Cl I
Percent of colorectal cancers with mutations in indicated genes
APC etc. >20%; hMSH2 etc. 10–20%; CNNBT etc. 3–10%
Methylation expression controlled genes
E-Cadherin, SFRP2,4,5, hMLH1, CDKN2A (p14, p16), CDX1.

**Table 2 cancers-15-02358-t002:** Tumor antigen targets for T cell-mediated killing.

Mutated Oncogenes:
Kras, TP53, Braf, APC, FBXW7, SMAD4 etc.
Differentiation antigens and over-expressed proteins
Her2, EGFR, CEA, MUC2, PSA, PSMA. WT1, gp100, tyrosinases.
Inappropriately expressed differentiation proteins
Glycosylation- e.g., abnormal glycosylation of MUC1, MUC2.
Phosphorylation
Viral antigens
e.g., from HPV (E7), HBV, EBV
Ectopic expression
AFP, PLAP.
Cancer testis antigens
NY-ESO-1, MageA3, etc.
Human endogenous retroviruses
HERVs (Human Endogenous Retroviruses)

## Data Availability

The authors declare no conflict of interest.

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
