# Peer review of "Cancer Immunotherapy: Where Next?"

_cancers, 2023, doi:10.3390/cancers15082358_

Round 1
Reviewer 1 Report
The review "Cancer Immunotherapy: Where next?" by Bodmer and Golubovskaya is a focused review of several current immunotherapies and their basis for efficacy. It is organized and detailed to an extent to give the reader a broad overview of relevant immunotherapies and due to the nature of this broad review is not highly detailed in any individual approach (which is fine). The link between evolutionary concepts and cancer immune responses is an interesting concept. The content should be of interest to the field and highlights emerging therapies including bispecific antibodies and T cell mimic antibodies. There are only minor spelling/formatting suggestions.
Author Response
- We thank reviewer for comments. We performed minor spelling andformatting revisions as suggested.
Reviewer 2 Report
Dr. Walter Bodmer and Dr. Vita Golubovskaya summarized the next step in cancer immunotherapy in a well concise manner in this review. This review is well written and major sectioned are covered.
Although major advancements in the field of immunotherapy are well discussed, a significant section is not highlighted. As a part of innate immune system Natural Killer (NK) cells are also a major player against cancer like the T cells. Major advancement has been made in the field of NK cell-based cancer immunotherapy. From immune checkpoint blocker antibodies and bi and tetra specific NK cell engager (Dr. Eric Vivier’s work) to CAR NK cells (Dr. Katy Rezvani’s wrok) are showing great success in clinical trial.
To make this review article up to date I will suggest to include a paragraph covering the NK cell based immune therapy part as discussed above.

Author Response
We would like to thank reviewer for comments. The reviewer suggested to include a paragraph covering the NK cell based immune therapy such as CAR-NK, NK-specific cell engagers and others.
We included a paragraph on NK cell-based immunotherapy as suggested by reviewer.
Reviewer 3 Report
The review article ‘Cancer Immunotherapy: Where next’ was well received. In this review, the authors have given a general introduction about different immunotherapeutic techniques that are being employed in different clinical settings or preclinical models for testing and treating different tumor types. The article seem to be well written and the language of the article is also acceptable. The paper gives a general introduction about the immunotherapies and at last has given short future perspective. The paper has no apparent flaw and can be accepted in current form
Author Response
We would like to thank reviewer for comments. We are glad that reviewer suggested to accept the review in the present form.
Round 2
Reviewer 2 Report
Dr. Walter Bodmer and Dr. Vita Golubovskaya summarized the next step in cancer immunotherapy in a well concise manner in this review. This review is well written and major sectioned are covered.
Although major advancements in the field of immunotherapy are well discussed, a significant section is not highlighted. Since topic of the review is - Cancer Immunotherapy: Where next?
I highly recommend including a full section after section-4 line no 161 to describe the below mention topic-
NK cell based immune checkpoint blocker antibodies, bi and tetra specific NK cell engager, CAR NK cells, on-going clinical trials involving them.
Author Response
Dr. Walter Bodmer and Dr. Vita Golubovskaya summarized the next step in cancer immunotherapy in a well concise manner in this review. This review is well written and major sectioned are covered.
Although major advancements in the field of immunotherapy are well discussed, a significant section is not highlighted. As a part of innate immune system Natural Killer (NK) cells are also a major player against cancer like the T cells. Major advancement has been made in the field of NK cell-based cancer immunotherapy. From immune checkpoint blocker antibodies and bi and tetra specific NK cell engager (Dr. Eric Vivier’s work) to CAR NK cells (Dr. Katy Rezvani’s wrok) are showing great success in clinical trial.
To make this review article up to date we have included a paragraph covering the NK cell based immune therapy part as discussed above.
Round 3
Reviewer 2 Report
I highly recommend to change the topic of the manuscript to – T cell based Cancer Immunotherapy: Where next?
And please remove all the previously added NK cell based part.
Author Response
We agree that we focused mainly on T cell therapies in the review. To be clearer to the reader we revised our review by the following:
- We included in the Introduction a sentence (lines 93,94) that while there are other immune system cells this review will focus mainly on T cell therapies.
The last Paragraph of Introduction is now read:
There are two fundamentally different ways to deploy the immune system. One is to increase the efficacy of the cancer patient's own immune system for effective attack on differences between the patient's cancer and normal cells. The second is to use antibodies, or T and other immune system cells, that are engineered to use the patient's immune system so that it attacks differences between normal and cancer tissues. This review will focus mainly on T cell-directed immune therapies.
1 We shortened a paragraph in Conclusion (lines 340-342) with an example of other immune cell therapies such as NK cell therapies to one sentence.
We would like also to highlight that in addition to T cell-based immunotherapies there are other immune cell type immunotherapies such as NK-cell based therapies, CAR-NK [27], and NK cell-specific engagers [28],[29],[30] that will be developed in future.
We think that this will be enough for publication.